# The Motivation Mechanism and Evolutionary Logic of Tourism Promoting Rural Revitalisation: Empirical Evidence from China

**Huizhan Wang [1], Kai Bai [2,\*], Lulu Pei [3], Xinru Lu [1] and Polish Mohanto [1]**

1   School of Management, Xi'an University of Science and Technology, Xi'an 710054, China
2   School of Geography and Tourism, Shaanxi Normal University, Xi'an 710119, China
3   School of Cultural Creativity and Tourism, Yuncheng Vocational and Technical University, Yuncheng 044000, China
\*   Correspondence: baikai@snnu.edu.cn

**Abstract:** It is difficult to ensure the sustainability of the practice of tourism promoting rural revitalisation, whether it is driven by a top-down exogenous power or a bottom-up endogenous power. Therefore, it is of great theoretical and practical significance to clarify the motivation mechanism and evolutionary logic of tourism promoting rural revitalisation. This paper first adopts a system theory research method to construct a model of the driving mechanism of tourism promoting rural revitalisation; then, it uses the Boston matrix model to construct an analysis matrix of the motivation mechanism of the promotion of rural revitalisation and uses this analysis matrix to explore the motivation mechanism and evolutionary logic of tourism promoting rural revitalisation under different development models. On this basis, the paper takes Yuanjia village (which lies in Liquan County, Shaanxi Province), a model village of tourism promoting rural revitalisation in China, as a typical case, and through the analysis of the practical process of tourism promoting rural revitalisation in this and the surrounding villages, it is verified that the logic and path of the evolution of the motivation mechanism from "exogenous power" to "endogenous power" is more suitable for the actual situation of tourism promoting rural revitalisation in China. The paper perhaps provides theoretical insights into and practical references for the practice of tourism promoting rural revitalisation in China and other developing countries in the world.

**Keywords:** rural tourism; rural revitalisation; motivation mechanism; evolutionary logic

## 1. Introduction

At present, with the changes in the main contradictions in Chinese society, the unbalanced development of the urban and rural areas and the inadequate development of the countryside have become the main constraints to meeting people's growing needs for a better life. To solve this new contradiction, the 19th National Congress of the Communist Party of China (CPC) proposed, for the first time in October 2017, the implementation of the rural revitalisation strategy, stressing the need to adhere to the priority development of agriculture and the rural areas and to accelerate the modernisation of agriculture and the rural areas in accordance with the general requirements of "thriving businesses, pleasant living environment, social etiquette and civility, effective governance, and prosperity" [1]. In October 2022, the 20th National Congress of the Communist Party of China (CPC) once again stressed the need to comprehensively promote the revitalisation of the countryside, solidly promote the revitalisation of the rural industries, talents, culture, ecology, and organisation, and accelerate the building of a strong agricultural industry in the country [2]. The revitalisation of the countryside entails vigorously developing the rural industries, taking multiple measures to enhance the vitality of agriculture and the rural areas, promoting industrial revitalisation, optimising the living environment, promoting rural civilisation, strengthening rural governance and improving the lives of farmers, making the countryside

gradually flourish and prosper, and ultimately achieving the goal of the comprehensive revitalisation of the countryside [3].

The Rural Revitalisation Strategy is a major national strategy implemented in China's rural areas to consolidate and expand the results of poverty eradication and accelerate the building of a strong agricultural country, based on precise poverty alleviation. It is very similar to the Rural Renaissance movement implemented in many developed countries, which was carried out against the backdrop of a declining and withering countryside. Since the 1970s, faced with the problem of rural decline in the process of rapid industrialisation and urbanisation, major countries around the world have actively implemented policy measures to promote rural reconstruction and revitalisation based on their national conditions [4]; these include the "Construction of Rural Centres" in the UK; the "Rural Revitalisation Plan" in France; the "Rural Competition Development Plan" in Germany; the "Comprehensive Demonstration Project for Villages and Towns" in Japan; and the "New Rural Movement" in South Korea [5–10].

The practice of rural revitalisation at home and abroad has sparked a great deal of academic interest in rebuilding and reviving the countryside. Within this interest, there are three main views on the research on the motivation mechanism of rural revitalisation: First, rural revitalisation is mainly driven by exogenous power. Based on the perspective of rural revitalisation, Chen Zanzhang explored the role played by the government in the integrated development of rural industries [11]; Zhang Hongyu explored the role played by entrepreneurs in the process of promoting the rural revitalisation strategy [12]. Second, rural revitalisation is mainly driven by endogenous power. Zhang Wenming et al. analysed the significance of endogenous development to rural revitalisation based on the three elements of resources, participation, and identity [13]; Liang Kunfei et al. analysed the endogenous construction path of rural revitalisation based on the rational perspective of the village community [14]; and Zhang Yuqiang et al. explored the motivation mechanism of the endogenous development of rural revitalisation using Shanghai Y village as an example [15]. Thirdly, rural revitalisation is driven by a combination of endogenous and exogenous power. From the perspective of the main body of rural cultural revitalisation, Gu Haiyan pointed out that the current rural cultural revitalisation needs exogenous power to stimulate endogenous power, but the generation of exogenous power has to be based on the premise and foundation of endogenous power [16]; Guo Zhen et al. explored the possible paths to achieving rural revitalisation based on the influence of internal and external social relationship networks on rural revitalisation [17]. The above-mentioned studies revealed that the power sources of rural revitalisation include mainly exogenous power and endogenous power and that the operation mechanism of rural revitalisation power has three forms, which are driven by exogenous power, driven by endogenous power, and driven by a combination of endogenous and exogenous power.

In the practice of rural revitalisation at home and abroad, tourism generally plays an important role. Because the practice of tourism promoting rural revitalisation is a relatively long process, tourism development models differ in terms of the role of the relevant stakeholders in rural revitalisation, and the corresponding motivation operating mechanisms also differ. In the government-led model, exogenous dynamics often play an important role in the practice of tourism promoting rural revitalisation. In the community-led model, endogenous dynamics often play an important role in the practice of tourism promoting rural revitalisation. In the "government-community" interlocking model, exogenous and endogenous dynamics interact with each other in the practice of tourism promoting rural revitalisation. Therefore, theoretically clarifying the evolutionary logic of the motivation mechanism of tourism promoting rural revitalisation can help guide the practice of tourism promoting rural revitalisation in a scientific manner.

This paper first explores the motivation mechanism and evolutionary logic of tourism promoting rural revitalisation under three different models: the government-led model, the community-led model, and the "government-community" interlocking model and, on this basis, takes Yuanjia village in Liquan County, Shaanxi Province, a model village of

tourism promoting rural revitalisation in China, as a typical case. Through the analysis of the practical process of tourism promoting rural revitalisation in Yuanjia village and its surrounding villages, it was verified that the evolutionary logic and path of the motivation mechanism from "exogenous power" to "endogenous power" is more suitable for the actual situation of tourism promoting rural revitalisation in China. This paper attempts to clarify in theory the motivation mechanism and its evolutionary logic of tourism promoting rural revitalisation under different tourism development models and to test in practice the operational characteristics of the motivation mechanism of tourism promoting rural revitalisation under different tourism development models, with a view to providing certain theoretical inspirations and practical references for the practice of tourism promoting rural revitalisation in developing countries in China and the world.

## 2. Literature Review

### 2.1. A Review of Research on Rural Revitalisation

The international research on rural revitalisation focuses on the following areas: The first comprises rural revitalisation policy formulation. The European Union and major member states have formulated diversified development policies to enhance the competitiveness of the countryside [18]. Germany has released a rural development and renewal plan [19]. The UK's rural construction takes empowerment as the main path [20]. The second comprises the exploration of rural revitalisation experiences. Pretterhofer H proposed the concept of rural urbanisation [21]. The United States launched the "three decent" rural reforms [22]. The third comprises the inspiration from rural revitalisation practices. South Korea and Japan attach importance to government guidance in their rural development practices and realise rural economic development methods that can highlight national cultural characteristics on the basis of full respect for the main position of the farmers [23].

Since China proposed the rural revitalisation strategy, rural revitalisation has attracted widespread attention from all walks of life and has become a high-frequency term in current Chinese academic research, with discussions on policies and theories emerging in a spurt [24]. Chinese scholars have conducted a great deal of research on the strategic significance, scientific connotation, target content, development mode, promotion path, impact effects, and many other aspects of rural revitalisation from the multi-dimensional perspectives of policy interpretation, regional development, the human–land relationship, urban–rural integration, and the practical experience summary [25,26] and have accumulated extremely fruitful research results in just a few years; these results have greatly promoted the goal recognition and theoretical cognition of rural revitalisation.

### 2.2. A Review of Research on Tourism Promoting Rural Revitalisation

In this process of rural revitalisation, rural tourism, as a diversified tourism activity closely linked to agricultural activities [27], has gradually become of concern to researchers. It is because the "economic, ecological, cultural, management and social benefits" generated by rural tourism are naturally coupled with the general requirements of rural revitalisation: "thriving businesses, pleasant living environment, social etiquette and civility, effective governance, and prosperity" (as shown in Figure 1). Among them, thriving businesses are the embodiment of the economic benefits of rural tourism and are also the focus of rural revitalisation. A pleasant living environment is the embodiment of the ecological benefits of rural tourism and is also the key to rural revitalisation. Social etiquette and civility are the embodiment of the cultural benefits of rural tourism and are also the guarantee of rural revitalisation. Effective governance is the embodiment of the management benefits of rural tourism and is also the foundation of rural revitalisation. Prosperity is the embodiment of the social benefits of rural tourism and is also the root of rural revitalisation [28].

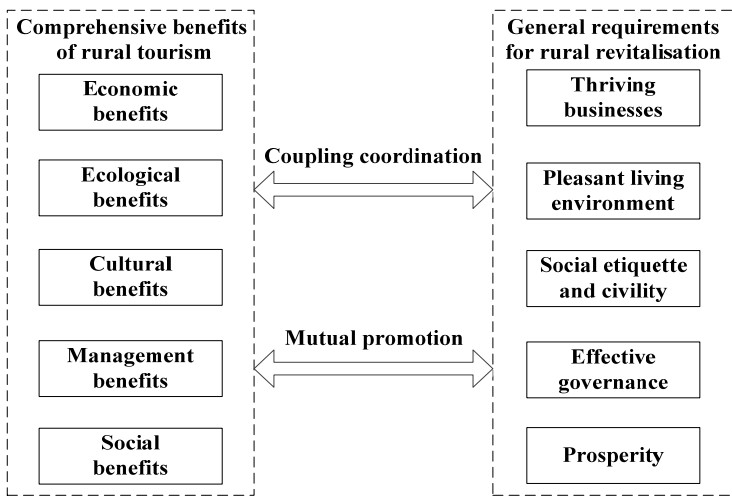

**Figure 1.** Coupling relationship model between the comprehensive benefits of rural tourism and the general requirements of rural revitalisation.

With the practical exploration of rural tourism, international scholars generally recognise the positive role of rural tourism in promoting rural rejuvenation and thus have begun to systematically study rural tourism as an important means of promoting rural development and achieving rural rejuvenation, with the research mainly focusing on the impact, mode, motivation mechanism, and development countermeasures of rural tourism development. After decades of research, a more comprehensive theoretical framework system has been created in terms of the promotion of rural development and revitalisation through rural tourism. China's research on tourism promoting rural revitalisation began with the introduction of the rural revitalisation strategy, and a series of achievements have been made, as previously explained, in terms of the impact, mode, mechanism, and path of tourism promoting rural revitalisation.

### 2.2.1. The Impact of Rural Tourism Development

International scholars have always paid considerable attention and attached great importance to research on the impact of rural tourism development. On the one hand, they have focused on the positive effects of rural tourism on rural socio-economic development, ecological environmental protection, and traditional cultural development. Gloria et al. argued that the development of rural tourism helps to promote the development of diversified local industries and the rapid development of the local economy [29]. Chen B pointed out that rural tourism contributes somewhat to the solution of rural water pollution problems [30]. Briedenhann et al. argued that the development of rural tourism helps to preserve rural cultural traditions [31]. On the other hand, concerns have been raised regarding the negative effects of the development of rural tourism. Gossling argued that, as tourists and tourism activities increase, the natural and socio-cultural atmosphere of rural tourism destinations will gradually fade, and noise and waste pollution will become increasingly problematic, ultimately leading to the degradation of the ecological environment and the distortion of the traditional culture in rural tourism destinations [32].

Chinese scholars generally recognise the pivotal role of rural tourism in the economic, social, political, ecological, and cultural aspects of the countryside. Through an analysis of the positive effects of rural tourism, Cai Kexin pointed out that rural tourism is conducive to the economic, cultural, social, and ecological revitalisation and governance transformation of villages [33]. Huang Zhenfang pointed out that moderate rural tourism is conducive to enhancing the cultural pride and identity of local community residents and preserving rural culture [34]. He Chengjun et al. stated that rural tourism development can promote the construction of beautiful villages [35]. Chinese scholars have also clearly recognised the negative effects that rural tourism has on the rural development process. Zhang

Gaojun et al. concluded that whether the promotion role of rural tourism development on rural revitalisation would change in the subsequent development process could not be confirmed [36]. Zhou Xing argued that, although rural tourism in concrete practice can bring back the local culture through a certain degree of folkloristic transformation, once the folkloristic cultural change is too commercialised and urbanised, it will weaken or even remove the characteristics and local flavor of the countryside [37], resulting in the unsustainable development of rural tourism and stalling the pace of rural revitalisation.

### 2.2.2. The Models of Tourism Promoting Rural Revitalisation

International scholars have summarised a variety of useful models for tourism promoting rural revitalisation based on the theoretical and practical studies of rural tourism. Simpson argued that the role of individual residents in rural tourism development is limited and that organisations or associations should be established to coordinate the interests of all parties, i.e., creating an 'individual + organization' model of development [38]. Walford argued that the business-led model is an important model for the development of rural tourism in a number of rural areas in England and Wales and that this model is beneficial to the economic development of rural tourism sites [39]. In their research on rural tourism, scholars such as Fallon and Reid have pointed out that community participation and benefits are prerequisites for the healthy and sustainable development of rural tourism [40,41].

Chinese scholars have classified and explored the models of tourism promoting rural revitalisation from various perspectives. According to the different subjects involved in rural tourism development, Xiang Fuhua categorised four models of tourism promoting rural revitalisation as "villagers (village collectives) build and operate themselves"; "villagers (village collectives) + tourists (citizens)"; "villagers (village collectives) + companies"; and "villagers (village collectives) + tourists (citizens) + companies" and considered the possibility that a village could adopt several models simultaneously in the development of tourism [42]. According to the different motivation mechanisms of rural tourism development, Wei Chao et al. proposed four types of tourism guidance models: the community enhancement model, the scenic area-dependent model, the cultural reconstruction model, and the suburban leisure model [43]. Based on a comparative summary of the rural tourism development experiences of Zhonghaoyu village in Shandong Province and Yuanjia village in Shaanxi Province, Wang Chenguang argued that the development of rural tourism in China at this stage should choose a "collective rural tourism development model" based on the basic system of the rural collective economy, with the village collective as the development link and all villagers' participation guided by development methods such as cooperatives, a joint-stock cooperative system, and a shareholding system model [44].

### 2.2.3. The Motivation Mechanism of Tourism Promoting Rural Revitalisation

The studies of international scholars on the motivation mechanism of tourism promoting rural revitalisation have mainly focused on tourism supply and demand. According to Walle, urban residents' interest in and desire for rural living environments directly contribute to the development and growth of rural tourism [45]. Maestro et al. argued that tourists choose rural tourism in order to explore the differences between urban and rural areas and to experience a different way of life in the countryside [46], while Francois et al. also analysed the needs of tourists in three major rural tourism destinations in France, including leisure and relaxation [47]. From the supply side, Streimikiene pointed out that rural tourism resources and their applications are important supply factors that drive the development of rural tourism [48].

The research of Chinese scholars on the motivation mechanism of tourism promoting rural revitalisation has mainly been carried out with regard to the following aspects. Yin Yuan et al. argued that rural tourism promoting the comprehensive implementation of the rural revitalisation strategy is a spiraling and hierarchical advancement process and that rural tourism needs to adjust its development status in terms of industrial positioning,

spatial development, host–guest communication, operation mechanisms, and economic benefits in accordance with the overall requirements of the rural revitalisation strategy to boost the comprehensive implementation of the rural revitalisation strategy [49]. Li Zhilong argued that the drive–response mechanism based on the participating subjects can promote the interaction and mutual influence of rural tourism and rural revitalisation and facilitate the operation, evolution, and development of the tourism promoting rural revitalisation system [24]. Wei Chao et al. pointed out that the tourism-led rural transformation and development in large urban fringe areas is the result of the combined effect of three major dynamics: the field force formed by the locational advantage, the cohesive force formed by the integration of internal forces in the countryside, and the exogenous force constituted by external inducing factors [43]. Cheng Ruifang et al. pointed out that, under the drive of rural revitalisation strategy, rural tourism development should take ecological environment monitoring and local culture protection as the guarantee mechanism [50]. Guo Jingfu et al. argued that tourism promoting rural revitalisation must be supported by certain institutional and regulatory innovations, and accordingly, they propose the development concept and mechanism of "pro-poor" growth, the institutional design of the "separation of three rights" of rural tourism resources and the institutional design of integrated urban and rural public services [51].

### 2.2.4. The Pathways of Tourism Promoting Rural Revitalisation

The research of international scholars on the countryside regarding tourism as a countermeasure to promote rural revival has mostly concentrated on summaries of practical experience in rural tourism. Duarte argued that the development of rural tourism in Portugal should focus on the marketing and promotion of rural tourism [52]. Using coffee plantations as an example, Apodacagonzdlez et al. suggested that coffee farms should be revitalised through the development of rural tourism with coffee experiences as a theme [53]. Roberta MacDonald, in a study based on the rural tourism industry in Canada, pointed out that the important factors and conditions for the development of rural tourism are policy support and characteristic rural culture [54]. With the continuous development of rural tourism, the relevant literature has gradually focused on the path and countermeasures of tourism promoting rural revival from the perspective of the positive effect of rural tourism development on rural revival [55–57].

The research of Chinese scholars on the paths of tourism promoting rural revitalisation has been based mainly on the general requirements of the rural revitalisation strategy. Dong Jing et al. advocated promoting the optimisation and upgrading of the rural tourism industrial structure by strengthening rural tourism market development and industrial management, changing marketing methods, improving infrastructure, and cultivating tourism talent to help rural areas to achieve comprehensive revitalisation [58]. Cai Xin-liang et al. pointed out that the traditional cultural resources of ethnic minorities should be creatively transformed and innovatively developed in the tourism industry through innovative ideas, creative approaches, holistic planning, branding, and system creation, in order to better play a role in promoting the implementation of the rural revitalisation strategy [59]. Based on the five dimensions of the rural revitalisation strategy, Song Huijuan et al. pointed out that the quality and efficiency of rural tourism can be improved by cultivating market subjects, creating a service-oriented government, promoting industrial integration, building a field and garden complex, and innovating development models to realise the "five-formed" development of rural tourism [60]. Huang Xiaojia et al. proposed four paths based on keeping the soul of the countryside, building a harmonious body, showing the beauty of the countryside and achieving a good job of "+ tourism" to build a tourism-oriented countryside and to promote the comprehensive implementation of the rural revitalisation strategy [61].

In summary, the relevant studies at home and abroad mainly cover the impact, mode, motivation mechanism, and path of the motivation mechanism of rural revitalisation, and the research contents are relatively comprehensive. Among them, in terms of the motiva-

tion mechanism, scholars at home and abroad have explained the motivation mechanism of tourism promoting rural revitalisation from different perspectives based on different research scenarios and cases. According to the above studies, in the Chinese context, the motivation mechanism of tourism promoting rural revitalisation can be mainly categorised into two types: from "exogenous power" to "endogenous power" and from "endogenous power" to "exogenous power". What is the evolutionary logic of these two paths? What are the results? The existing studies do not provide sufficient explanations. Only by theoretically clarifying the evolutionary logic of these two paths can the results be predicted, and then, a suitable path for the evolution of the motivation mechanism of rural revitalisation in China can be constructed.

*2.3. A Review of Research Methods*

At present, the research on tourism promoting rural revitalisation is basically based on qualitative research, and case studies and inductive methods have become the mainstream of the research, such as in MacDonald's research on the rural tourism industry in Canada [54], Simpson's research on the rural tourism development model [38], Guo Jingfu's research on the system and path of tourism promoting rural revitalisation in ethnic areas [51], and Lu Junyang's research study on the realisation mechanism and social support of rural tourism contributing to rural revitalisation [62]. Relatively speaking, mathematical and statistical methods are lacking, and the systematic, quantitative, and standardised theoretical research and quantitative research results are relatively scarce.

## 3. Research Methodology

This paper first constructs a model of the motivation mechanism of tourism promoting rural revitalisation based on the stakeholder theory and the system theory research method; then, it uses the Boston matrix model to construct an analytical matrix of the motivation mechanism of tourism promoting rural revitalisation. Using this analytical matrix, the paper explores the motivation mechanism and evolutionary logic of tourism promoting rural revitalisation under the government-led model, the community-led model and the "government-community" interlocking model. Finally, the paper uses a case study approach to analyse the promoting mechanism and evolutionary logic of tourism promoting rural revitalisation in Yuanjia village and its surrounding villages; Yuanjia village is a model village for tourism promoting rural revitalisation in China.

## 4. Research Results and Analysis

*4.1. The Motivation Mechanism and Evolutionary Logic of Tourism Promoting Rural Revitalisation*

4.1.1. The Motivation Mechanism of Tourism Promoting Rural Revitalisation

Tourism promoting rural revitalisation is a complex systemic project [63], involving community residents, local governments, tourism enterprises, tourists, NGOs, and other relevant interest subjects in the development and evolution [64]; each of these has different interests in the practice of tourism promoting rural revitalisation, and the way they play their roles varies. Among them, community residents, as the owners of the villages, have the main interest demands with regard to participation in rural tourism development, such as achieving local employment, improving tourism management and service skills, increasing income, and improving family living standards [65]. The research has confirmed that the greater the development opportunities and the benefits that community residents receive from rural tourism development, the more motivated they are to participate in rural tourism [66]. Therefore, community residents are the main endogenous driving force for rural tourism development and rural revitalisation and contribute to the practice of rural revitalisation through active, comprehensive, and in-depth participation in rural tourism development [67]. As participants in rural tourism, the main interests of local governments, tourism enterprises, tourists, and NGOs in participating in rural tourism development are to obtain economic or political returns [68], to obtain tourism benefits [69], to seek pleasure

and experience [70], to promote community participation, and to protect the ecological environment [71], respectively, through policy support [72], capital investment [73], tourism consumption [74], supervision and management [75], etc.; these are the exogenous drivers of rural tourism development and rural revitalisation.

According to materialistic dialectics, the development of things cannot be separated from the joint action of internal and external factors. Internal factors are the basis for the change and development of things, external factors are the conditions for the change and development of things, and external factors work through internal factors [76]. Obviously, in the practice of tourism promoting rural revitalisation, there is a dialectical and unified relationship between endogenous and exogenous power. Among them, endogenous power is the fundamental force of tourism promoting rural revitalisation, and only when rural residents have the right, opportunity, and willingness to participate in rural tourism development can rural tourism achieve sustainable development; exogenous power is the supporting force for tourism promoting rural revitalisation, and only with the strong support of external interested parties, such as the government, tourism enterprises, tourists, and NGOs, in terms of policies, funds, markets, and supervision, can rural tourism achieve rapid development. This can occur only when the endogenous and exogenous powers achieve a positive interaction, i.e., the community residents actively participate in the development of rural tourism, which draws the attention of external interests to rural tourism and the external interests further stimulate the community residents' enthusiasm to participate in tourism through support in terms of policies, funds, markets, and regulation, thus promoting the rural revitalisation. The practice of tourism promoting rural revitalisation further promotes the deep and positive interaction between endogenous and exogenous power (as shown in Figure 2).

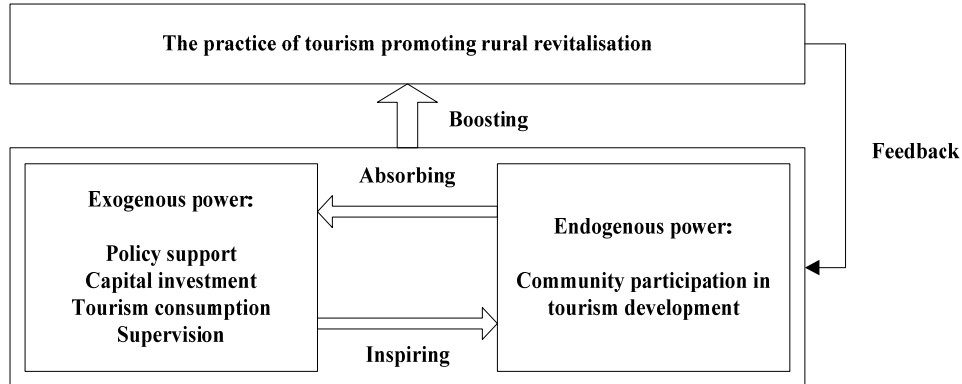

**Figure 2.** The model of the motivation mechanism of tourism promoting rural revitalisation.

In the practice of tourism promoting rural revitalisation in China, due to the differences in the foundation, development stage, and mode of tourism, the depth and breadth of the participation of the relevant stakeholders in the practice of tourism promoting rural revitalisation also vary, and thus the strength and combination of the endogenous and exogenous powers differ. Based on the strengths and weaknesses of the endogenous and exogenous powers in the practice of tourism promoting rural revitalisation in China, we use the Boston matrix model to construct an analysis matrix of the motivation mechanism of tourism promoting rural revitalisation, as shown in Figure 3.

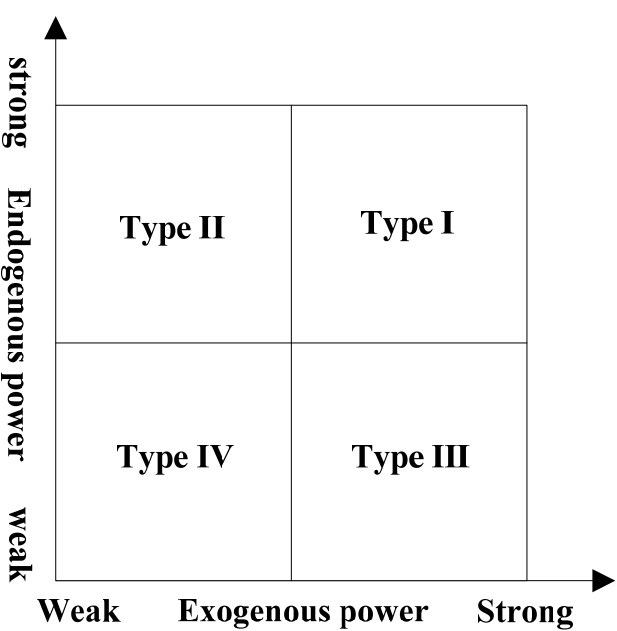

**Figure 3.** Analytical matrix of the motivation mechanism of tourism promoting rural revitalisation.

As shown in Figure 3, in the practice of tourism promoting rural revitalisation, the combination of endogenous and exogenous power is divided into four types, as follows.

Type I: Strong endogenous power and strong exogenous power. The endogenous power and the exogenous power achieve a benign interaction in the practice of tourism promoting rural revitalisation, with rural tourism developing continuously and rapidly and the countryside gradually achieving comprehensive revitalisation. At present, a small number of national or provincial rural revitalisation demonstration sites in China belong to this type.

Type II: Strong endogenous power and weak exogenous power. The development of rural tourism mainly relies on endogenous power, while the role of exogenous power is limited. The process of tourism promoting rural revitalisation is relatively long, and the development of rural areas may become stagnant or even decline due to the lack of necessary external support. Generally, villages with a "community-led" tourism development model fall into this category.

Type III: Weak endogenous power and strong exogenous power. Rural tourism development relies mainly on exogenous power, and the role of endogenous power is limited. With strong external support, rural tourism may achieve rapid development in the short term. However, as it is difficult to effectively stimulate the endogenous motivation of community residents in the short term, the development of tourism in such villages may not be sustainable; thus, rural development becomes somewhat uncertain. Many of the villages in China that have implemented a strategy to alleviate poverty through tourism during the 13th Five-Year Plan fall into this category.

Type IV: Weak endogenous power and weak exogenous power. With insufficient endogenous motivation and limited input from external sources, villages tend to fall into decline. At present, a large number of "hollow villages" in China fall into this category.

In the Chinese context, according to the initial state of the endogenous power and the exogenous power of rural tourism development and their evolution, the practice of tourism promoting rural revitalisation has two possible outcomes: "revitalisation" and "decline". Among them, the "revitalisation" path tends to be the development path according to Type I. Rural tourism continues to develop along the evolutionary path of "II→I" or "III→I" and eventually evolves into prosperous countryside. The "decline" path is the development path according to type IV, where the countryside gradually falls into a state of "stagnation"

along the evolutionary path of "II → IV" or "III → IV" and eventually evolves into a declining countryside.

Theoretically, the motivation mechanism and the evolutionary logic of tourism promoting rural revitalisation are closely related to the mode of tourism promoting rural revitalisation. In practice, there are mainly two models of tourism promoting rural revitalisation in China: "government-led" and "community-led". Among them, the "government-led" model emphasises a top-down exogenous impetus, while the "community-led" model places emphasis on a bottom-up endogenous impetus. Therefore, the motivation mechanism and the evolutionary logic of tourism promoting rural revitalisation under the two models are fundamentally different.

### 4.1.2. The Motivation Mechanism and Evolutionary Logic of Tourism Promoting Rural Revitalisation under the Government-Led Model

At present, most of the villages in China that mainly rely on tourism promoting rural revitalisation adopt the "government-led" model. This is because most of these villages used to be poor villages, and they only started to move out of poverty through tourism development according to China's 13th Five-Year Plan; so, they are still at the primary stage of tourism development. The community residents of these villages have not yet been effectively stimulated to develop rural tourism; the ability to develop themselves is still relatively weak, and tourism development mainly relies on a variety of exogenous impetuses. Among them, the local government plays a leading role in the practice of tourism promoting rural revitalisation [62]. On the one hand, the local governments use their political power to directly promote the development of rural tourism through policy support, financial investment, education and training, etc. On the other hand, the local governments provide conditions for other external stakeholders to participate in rural tourism development through their strong mobilisation and organisational capabilities. For example, they inject funds and bring advanced management experience to rural tourism development by attracting investment; by introducing preferential policies to stimulate rural tourism consumption; and by creating a policy environment to facilitate the participation of NGOs in rural tourism development.

There is no doubt that the government-led model is relatively effective in the early stages of rural tourism development as it is able to focus the factors of production, such as capital, labor, and technology, on the countryside in the short term, thus enabling rural tourism to start as quickly as possible. China's accomplishment of the daunting task of poverty eradication during the 13th Five-Year Plan is ample proof of the efficiency of the government-led model in the short term. However, the realisation of rural revitalisation is a long-term process. With the successful completion of China's poverty eradication task, which caused enormous financial pressure, the financial, material, and human resources invested in rural tourism by governments at all levels during the rural revitalisation phase are bound to gradually decrease compared to the precise poverty alleviation phase. Under such circumstances, if the countryside relies on government assistance and the endogenous power of developing rural tourism is not effectively stimulated, it will be difficult to sustain the development of rural tourism, and the countryside will not only have difficulty in achieving revitalisation but may also decline.

In conjunction with Figure 4, in the government-led model, local governments, in pursuit of "visible changes", i.e., the so-called "performance projects", are more likely to focus on hardware construction, such as improving tourism infrastructure and optimising the ecological environment, while not enough attention is paid to the "invisible changes", such as institutional construction, education and training, cultivating wisdom, and promoting encouraging activities. If hardware construction is the "engine" of rural tourism development, then software construction is the "fuel" of rural tourism development. Without the heat of the fuel to push, the engine will not work. The government-led model can often lead to the "superficial" revitalisation of villages in the early stages of tourism development along the "IV → III" evolutionary trajectory in the short term, but this model further

reinforces China's "strong government, weak society", which is prone to path dependence and lock-in effects [77], resulting in community residents being increasingly marginalised in the process of developing rural tourism and gradually losing the initiative to participate in tourism development and rural revitalisation. Once the exogenous impetus is weakened, community residents are likely to cause rural tourism sites to decline along the "III → IV" evolutionary path due to the lack of endogenous impetus and self-development capacity.

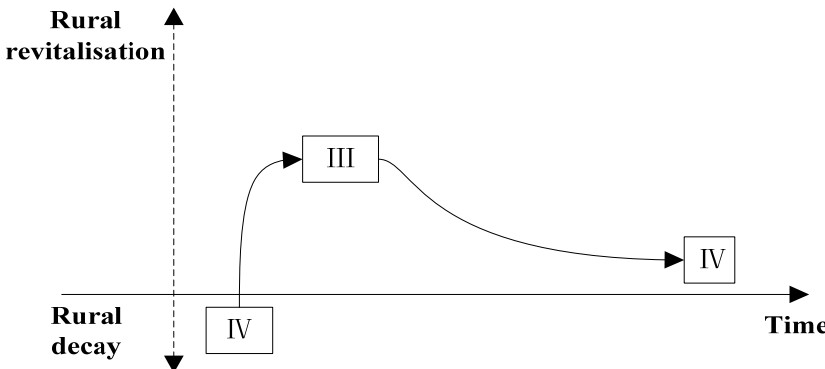

**Figure 4.** Evolutionary logic of the motivation mechanism of tourism promoting rural revitalisation under the government-led model.

4.1.3. The Motivation Mechanism and Evolutionary Logic of Tourism Promoting Rural Revitalisation under the Community-Led Model

At present, in the practice of tourism promoting rural revitalisation in China, only a very small number of villages have formed a "community-led" model based on villagers' autonomy and guaranteed by the villagers' shared governance. These villages often have a good foundation and conditions for tourism development, and rural tourism usually originates from the demonstration of a small number of capable community members, rather than from the top-down administrative drive of the local government. In the Chinese context, there is often a certain degree of uncertainty about the direction of the development of this type of tourism promoting rural revitalisation.

Combined with Figure 5, in the early stages of rural tourism development, due to the relatively small number of community residents spontaneously participating in rural tourism, the endogenous power of the community to develop rural tourism on their own has not been fully and deeply stimulated; this is coupled with the relatively limited support and assistance provided by external stakeholders, resulting in rural tourism facing major difficulties in sustainable development, such as disorderly development, lack of funds, and talent shortage. If these difficulties are not fundamentally resolved within a long period of time, the development of rural tourism will be slow and ineffective, and it will be difficult for the majority of community residents who are on the fence to participate in rural tourism, thus causing rural tourism to fall into a decline along the evolutionary trajectory of "IV→II→IV", and the role of tourism in rural revitalisation will diminish.

Theoretically, if the above difficulties are effectively overcome, i.e., with the development of rural tourism, driven by the demonstrators, the endogenous motivation of community residents to develop tourism is gradually stimulated, the capacity for self-development is gradually improved, more and more community residents commit themselves to the development of rural tourism, rural tourism follows the "IV→II→I" evolutionary path to achieve prosperity, and the role of tourism for rural revitalisation will become increasingly prominent. However, in the Chinese context, it is difficult for rural tourism to achieve sustainable development simply by relying on the spontaneous power of communities. This is because the property rights of rural tourism resources in China are usually owned by several government departments at the same time, and the complexity of the property rights of rural tourism resources determines the inevitability of the "tragedy of the commons" [78], which makes rural villages face many externalities in the process of developing tourism,

such as, typically, the "tourism enclaves" and "hitchhiking". These problems require the use of external forces to provide compulsory institutional arrangements and designs [79] before they can be fundamentally alleviated, while the community-led model based on the internal forces of the community can hardly resolve the "tragedy of the commons" fundamentally. After the development of rural tourism has reached a certain level, it will gradually decline along the path of "I→II→IV", and the boosting effect of tourism on rural revitalisation will also lack sustainability.

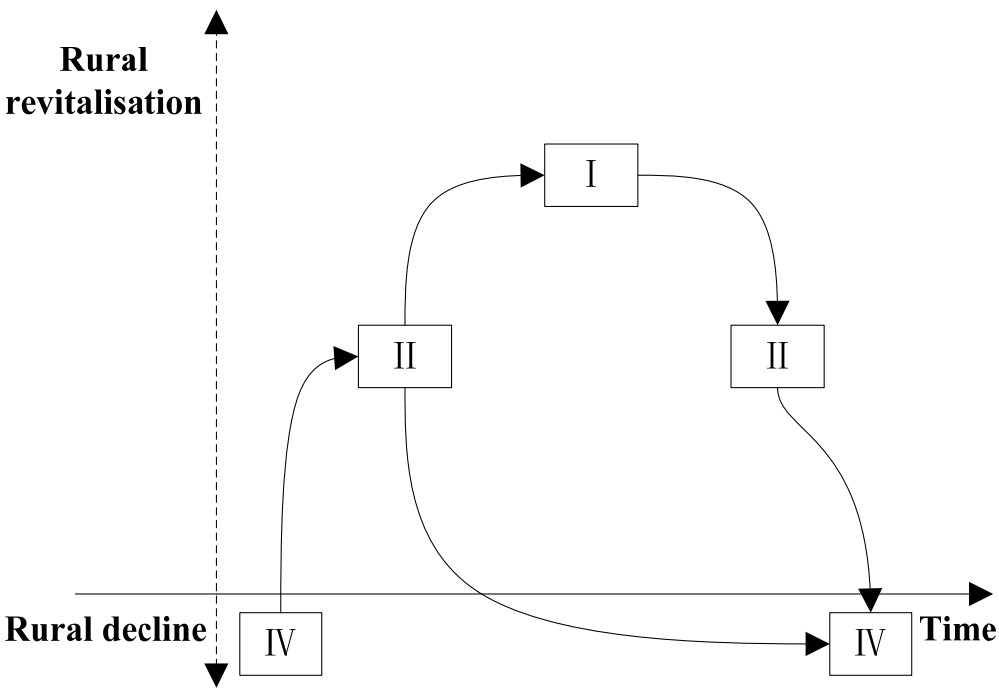

**Figure 5.** Evolutionary logic of the motivation mechanism of tourism promoting rural revitalisation under the community-led model.

4.1.4. The Motivation Mechanism and Evolutionary Logic of Tourism Promoting Rural Revitalisation under the "Government–Community" Interlocking Model

In the practice of tourism promoting rural revitalisation, the government-led and community-led models have advantages and disadvantages. Under the government-led model, there is a relatively abundant external source of motivation, which can provide the necessary policy support, financial investment, and education and training for the development of rural tourism, thus helping rural tourism to achieve rapid development in the short term. However, under this model, rural residents are more passive participants in tourism development, with insufficient endogenous motivation to develop tourism, and the phenomenon of spiritual poverty is not effectively resolved. The problem of "waiting and relying on others" and path dependence is serious. Once the support from external interests is reduced, it will be difficult for rural tourism to achieve sustainable development, and its role in promoting rural revitalisation will be greatly affected. Under the community-led model, rural tourism can be developed to a certain extent by relying on the demonstration and driving role of capable community members. However, under this model, it is difficult to avoid the problems of the "tragedy of the commons" and "free-riding", making it difficult to achieve sustainable development.

In summary, the problems of spiritual poverty and path dependency faced by rural tourism development under the government-led model are mainly due to the lack of endogenous motivation of community residents to participate in tourism, but these problems can be effectively solved under the community-led model; the problems of the tragedy of the commons and "tourism enclaves" faced by rural tourism development under the community-led model are mainly related to the design of the government's institutional

arrangements, but these problems can be effectively alleviated under the government-led model. Based on this, this study attempted to combine the above two models and build a third model that combines the advantages of both the government-led and the community-led models: the "government–community" interlocking model. Under this model, the organic combination of government administration and community self-governance can effectively stimulate the synergy of the endogenous and exogenous forces in the development of rural tourism and promote the sustainable development of rural tourism and thus sustainably promote rural revitalisation.

In the context of China's rural revitalisation strategy, in order to achieve the comprehensive revitalisation of the countryside on schedule, it is necessary to adopt a government-led model in the early stages of rural tourism development for the villages that have certain tourism development foundations and conditions and rely mainly on tourism promoting rural revitalisation. However, it is important to note that, while the government arranges other external stakeholders to inject considerable external impetus into rural tourism development, it should also focus on stimulating the endogenous impetus of community residents to participate in rural tourism development and enhance their tourism development capacity and gradually change its role from being a "dominant" to a "service provider" as rural tourism develops. The development of rural tourism will gradually change its role from "dominant" to "servant" [80] and through gradual tourism empowerment the initiative of tourism development and rural construction will be passed on to community residents, thus gradually transforming the rural tourism development model from "government-led" to "community-led". Thus, the momentum of rural tourism development is gradually transformed from "exogenous momentum" to "endogenous momentum" so that the development of rural tourism from "blood transfusion" development to "blood-building" development will occur. This will enable the countryside to start and develop as soon as possible with sufficient external impetus in the early stages of tourism development, but also to achieve the sustainable development of rural tourism during the development and prosperity of rural tourism, due to the full stimulation of endogenous impetus for the development of tourism by community residents and the significant increase in the self-development capacity. At the same time, it helps to reduce the workload of local governments to a certain extent so that they can shift more energy to other regions that are in the primary stages of development [72] to optimise the allocation of exogenous resources and improve the efficiency of rural revitalisation.

In the context of Figure 6, the government-led model is the common choice for most rural villages in the early stages of tourism development. With the rapid start and development of rural tourism, the shift from "government-led" to "community-led" has become the inevitable choice for most villages. Therefore, along the path of "IV→III→II→I", the sustainable development of rural tourism is gradually achieved, thus promoting the overall revitalisation of the countryside.

However, there are also a few villages in China that are rich in tourism resources and developed tourism earlier, where rural tourism is mainly driven by the demonstration of capable community members and relies mainly on the endogenous power of the community to start and develop. In the middle and late stages of rural tourism development, with the gradual intervention of the local government, and with the joint efforts of the community and the government, the sustainable development of rural tourism can also be achieved, thus promoting the overall revitalisation of the countryside. Specifically, in the early stages of tourism development, due to the social influence of the community's competent people, it is easier to attract the attention of external interested parties other than villages [17] and thus to obtain the external resources necessary for the development of rural tourism, such as policy support, financial investment, and education and training, to promote the rapid development of rural tourism. The rapid development of rural tourism, in turn, helps to attract more community residents to participate in tourism development and, through rural tourism practices, continuously stimulates the endogenous motivation of community residents and enhances their capacity for tourism development. However, it is important

to note that, in the Chinese context, once exogenous forces have become involved in rural tourism development, they are usually more powerful and have absolute decision-making power [81], thus making such rural tourism development models evolve from "community-led" to "government-led". This has led to the gradual evolution from a "community-led" to a "government-led" model of rural tourism development. However, such villages, as a result of the primary stage of tourism development, mainly rely on the internal strength of the community to obtain a certain degree of development; therefore, a community with external interested parties has a certain amount of negotiating capital, which makes the community able to obtain external and incremental tourism power relatively easily. The tourism development model eventually evolves into the "government–community" interlocking model. Under the cooperative game between the community and the external interest subjects, along the path of "IV→II→III→II→I", the sustainable development of rural tourism is gradually achieved, thus promoting the comprehensive revitalisation of the countryside.

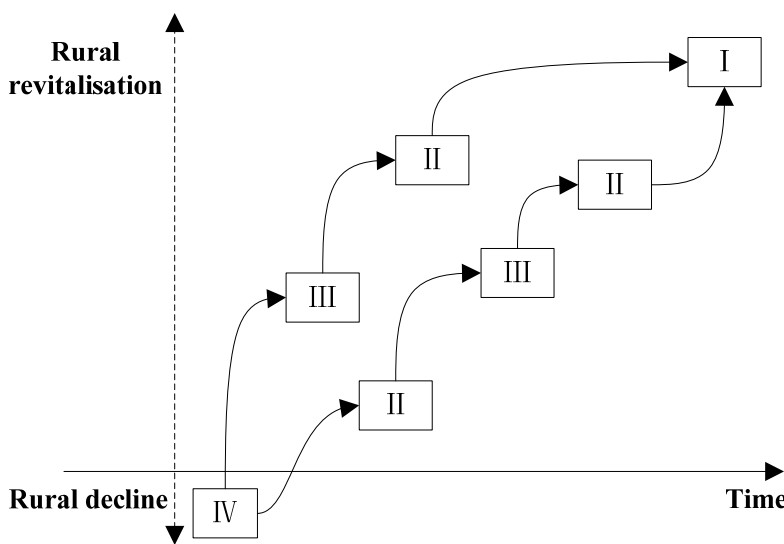

**Figure 6.** Evolutionary logic of the motivation mechanism of tourism promoting rural revitalisation under the "government-community" interlocking model.

This path is not representative in the Chinese context, but it has certain practical implications. The revelation is that, although communities are the leading force in rural tourism development and rural revitalisation, it is difficult for them to achieve full rural revitalisation alone as expected within the national strategic timeframe. Therefore, in the early stages of tourism development, the involvement of external forces is both necessary and essential. However, the involvement of external forces often places the community in a subordinate position in tourism development, which requires the community to strengthen the game with external interests in rural tourism development. This is because the power of the community residents is realised by them through a continuous and iterative process of gaming with other stakeholders [82]. Only through tourism empowerment in a continuous game can they once again gain a dominant position in rural tourism development and thus promote the sustainable development of rural tourism and promote comprehensive rural revitalisation.

In summary, both the government-led and the community-led models can evolve into "government–community" interlocking models through certain institutional arrangements. However, in the Chinese context, the development model from "government-led" to "community-led" and from "exogenous power" to "endogenous power" is obviously more in line with the reality of China's tourism in promoting rural revitalisation. From "government-led" to "community-led" and from "exogenous power" to "endogenous power", for local governments, means a return to a limited government from an unlimited

government, and for the community, it means a change from passive development to active development. It is important to note, however, that after the return of the local government from an unlimited to a limited government, even if there are various agents (such as community-based organisations and tourism enterprises) that take over the management of rural tourism development, the local government still has to play a necessary role in guiding or directing the development of rural tourism. This is because, in rural tourism development, the principal agent problems are often accompanied by "moral hazard" and "adverse selection" [83], and rural tourism may stagnate or even regress if the agents do not fulfil their responsibilities [36]. Therefore, "community-led, government participation" should be the ideal state of the "government–community" interlocking model, which is also the appropriate model for tourism promoting rural revitalisation in the Chinese context, and the corresponding motivation mechanism change from "exogenous power" to "endogenous power". This is the general path for the evolution of the motivation mechanism of tourism promoting rural revitalisation in the Chinese context.

*4.2. Case Presentation: The Motivation Mechanism and Evolutionary Logic of Tourism Promoting Rural Revitalisation of Yuanjia Village in China*

4.2.1. Typicality of the Case

Yuanjia village is located in Liquan County, Shaanxi Province, China. From the 1970s to the present, Yuanjia village has undergone three stages of development, agricultural stabilisation, industrial enrichment, and tourism development (as shown in Figure 7), achieving a transformation from "poor countryside" to "beautiful village" and becoming a model of rural revitalisation in Shaanxi Province and China.

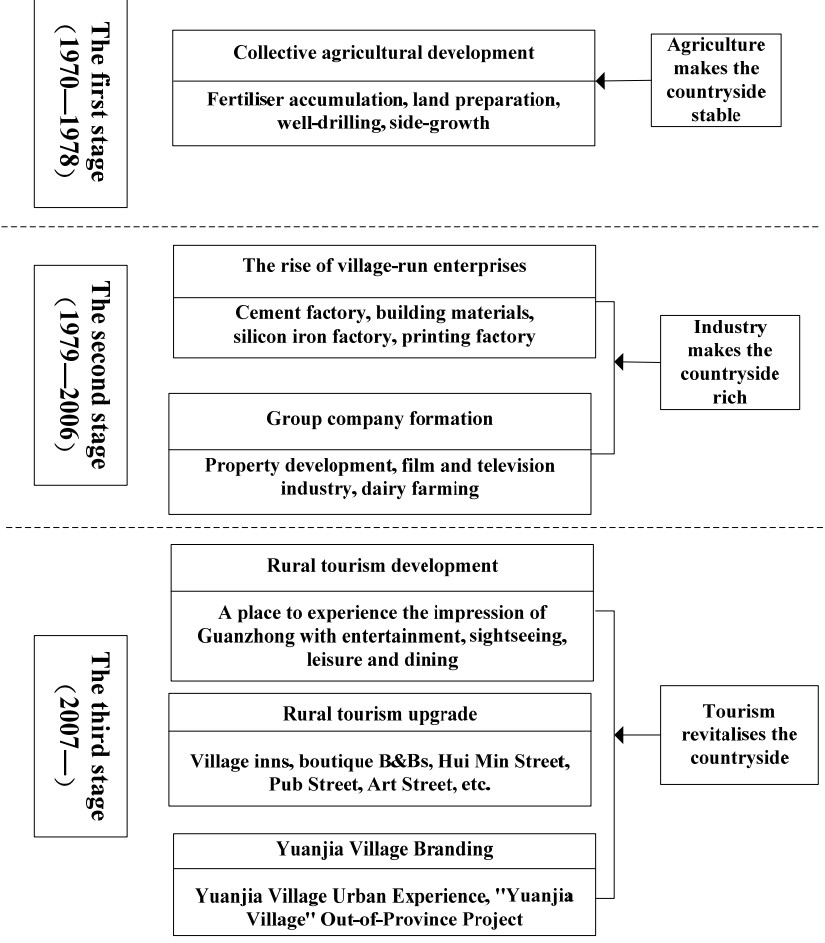

**Figure 7.** The development history of Yuanjia village.

In 2019, Yuanjia village, with a household population of just over 200, received more than 5.8 million visitors, with a total tourism income of more than CNY 1 billion and an annual per capita income of more than CNY 100,000 for the community residents [84]. In the practice of tourism promoting rural revitalisation, Yuanjia village has not only directly driven its own villagers out of poverty and into wealth but has also enabled the surrounding villages to benefit from its rural tourism development through the development strategy of "one village leading ten villages", which has to a certain extent promoted the economic and social development of the surrounding area. In 2019, Yuanjia village employed more than 3,000 people, leading more than 20,000 farmers in the surrounding area to increase their income [85]. This paper used Yuanjia village as an example and verified the rationality of the evolutionary path of the motivation mechanism of tourism promoting rural revitalisation from "exogenous power" to "endogenous power" by analysing this evolutionary process in Yuanjia village and the surrounding villages.

### 4.2.2. Data Acquisition

The research team conducted a pre-study in Yuanjia village on 6 June 2020, using a snowball sampling method to conduct one-on-one interviews with some village cadres and community residents, resulting in an 11,000-word interview text. During this process, theoretical sampling and semi-structured interviews were used, with an average interview time of about 35 minutes, covering basic information about individuals and families, the development process of rural tourism, the impact of rural tourism development on individuals and communities, and its evolutionary process. On the basis of the interviewees' knowledge and consent, a combination of on-site recording and on-site audio recording was used to record the interviews. A total of 34 valid interviews were eventually collected, and a total of about 94,000 words of interview notes were compiled. The researchers used a variety of data collection methods in a natural prospective to explore the social phenomenon of case-based tourism promoting rural revitalisation in a holistic manner, using an inductive approach to analyse the data and develop theories and gaining an interpretive understanding of the behaviour and construction of meaning through interaction with the research participants.

### 4.2.3. Motivation Mechanism and Evolutionary Logic of Tourism Promoting Rural Revitalisation of Yuanjia Village

(1) From "Endogenous Power" to "Exogenous Power": Yuanjia Village's Practice

As a pioneer in tourism promoting rural revitalisation, Yuanjia village has successfully explored a new path of rural collective economic development in a new era of socialism with Chinese characteristics. By looking back at the history of Yuanjia village's rural economic development, we find that its current model of rural tourism development, which is "led by the village collective and dominated by the villagers", has a distinct historical imprint and a solid practical foundation. From the 1970s, when the village focused on agricultural production, to the 1980s and 1990s, when the village focused on industrial development, Yuanjia has always followed the development idea of "not wavering in the collective economy, relying on agriculture to stabilise the village and industry to enrich the village", and has vigorously developed the village collective economy, which has grown from scratch and from weak to strong. Although the village's collective economy has been severely affected by national policy adjustments in the course of its development, the market awareness and business skills developed by the village cadres and the majority of villagers through their long involvement in collective economic activities have enabled Yuanjia village to grasp opportunities and transform itself in time when faced with external threats and opportunities. In 2007, against the backdrop of the accelerated development of rural tourism in China, Yuanjia village was able to rise to the occasion. The village's development entered the stage of "tourism revitalising the countryside", and the role of tourism in the revitalisation of the countryside began to emerge gradually. Specifically,

the evolutionary process of Yuanjia village's tourism promoting rural revitalisation can be divided into the following three stages.

The first stage: the Grassroots Organisation-led Stage. In 2007, in response to the increasingly prominent ecological and environmental problems brought about by industrialisation, the CPC proposed for the first time at the 17th National People's Congress that "building an ecological civilisation should be a new requirement for achieving the goal of building a moderately prosperous society in all aspects", calling for a new level of understanding and resolution of the resource and environmental issues from the perspective of the progress of civilisation and for a comprehensive review and response to the serious challenges faced by humans and nature, the environment, and the economy in the economic, political, cultural, social, and scientific fields. In this context, most of the collective enterprises in Yuanjia village were ordered to be rectified or transformed due to their high-energy consumption, high emissions, and high pollution. At a critical time in the development of Yuanjia village, Guo Zhanwu, then Secretary of the Party Branch of Yuanjia village, analysed the situation and, using Yuanjia village's rich Guanzhong folklore resources, mobilised the community to tap into the local folklore culture and develop rural tourism. Under the leadership of Guo Zhanwu and the grassroots organisation of Yuanjia village, the residents of the community began to gradually participate in the development of tourism, using their own houses and courtyards to set up farm caravans; at the same time, the grassroots organisation introduced skilled craftsmen from the surrounding villages to run the workshops through a variety of support and assistance measures. However, as rural tourism was new for most of the community residents, at the beginning of the development of rural tourism, even with the demonstration of community elites and the support of grassroots organisations, only a small number of community residents in Yuanjia village responded to the call to take the lead in tourism activities, while most community residents remained on the sidelines and were hesitant about tourism development. However, with the gradual increase in the number of tourists, the community residents who participated in the tourism development first gained a relatively substantial economic income, and as a result, most of the community residents who were on the sidelines began to actively participate, renovating their own homes and operating farmhouses in accordance with the law.

In summary, Yuanjia village has gradually worked out a development model in which the grassroots organisation is the leading core and the community residents are the main participants in the development of rural tourism in its initial stage. With the gradual increase in the number of community residents participating in tourism, the scale of Yuanjia village's farmhouses continues to expand, increasing the residents' incomes and changing their production and lifestyle and ideology, while also promoting the improvement of the village's appearance and infrastructure construction.

The second stage: the Individual Independent Development Stage. With the rapid development of rural tourism, the relatively undiversified form of tourism supply in Yuanjia village was no longer able to meet the diversified needs of the tourism market. In this context, Yuanjia village developed a snack street and a workshop street on the basis of the original farmhouses and set up co-operatives one after another, relocating the original advantageous industrial projects of the old street to the workshop street by way of capital increase and shareholding and promoting the newly developed projects in the form of shareholding and dividends. Driven by the economic benefits of tourism, the residents in the vicinity chose to be employed in tourism and the related industries nearby.

With the continuous development of the snack street and workshop street, Yuanjia village has gained fame and attracted the attention of Guanzhong folk culture, with more and more tourists flocking to Yuanjia village, creating considerable employment and entrepreneurial opportunities for the local residents. In order to extend the tourism consumption chain and increase the length of stay of the tourists, Yuanjia village continues to transform and develop rural holiday tours; it promotes the construction of night tourism projects and develops tourism cultural and creative products, which not only further stim-

ulates the enthusiasm of local residents to return to their hometowns to start their own businesses, but also attracts a large number of foreign residents to come to the village to start their own businesses. Yuanjia village is thus gradually achieving a flourishing development of related industries, such as catering, accommodation, sightseeing and entertainment, and tourism shopping. With the boost in rural tourism, the process of revitalising the countryside in Yuanjia village has begun to accelerate, with significant improvements in industry, organisation, talent, culture, and ecology.

To sum up, as the demand for rural tourism becomes high-end and diversified, and in order to obtain more income from tourism development, the residents of Yuanjia village continue to improve their awareness of self-development and to improve their self-development abilities, gradually becoming the main force of Yuanjia village's tourism promoting rural revitalisation. During this period, Yuanjia village adjusted its industrial structure and initially opened up the supply chain between the third, second, and first industries on the basis of the development of tourism. The transformation from "attracting tourists" to "retaining tourists" has been completed, and the number of tourism workers has been further increased and their scope further expanded.

The third stage: the Joint Promotion by Government and Enterprises. As the popularity and influence of tourism in Yuanjia village increased, the government at the higher levels began to use Yuanjia village as a model for rural tourism development, gradually increasing its management and support for the tourism industry and adopting a series of strong policy measures to promote the sustainable development of tourism in Yuanjia village. On the basis of respecting the leading position of Yuanjia village's grassroots organisations in tourism development, the higher government accurately understood its own position and provided support for Yuanjia village's tourism development in terms of infrastructure construction, land approval, project approval, market supervision, and tourism publicity, which injected a strong exogenous impetus for the transformation and upgrading of Yuanjia village tourism and comprehensively contributed to the revitalisation of the countryside.

Yuanjia village, with its expanding tourism market and continuously improving investment environment, has attracted a large amount of external capital or enterprises to accelerate their investment in tourism, but the development of tourism in Yuanjia village is rooted in the local vernacular culture and is closely related to rural production and life. In order to avoid the negative externalities introduced by capital to the countryside and to prevent the individual and collective interests of the community residents from being compromised, the Yuanjia village's grassroots government has defined the "threshold" for external tourism enterprises to intervene, requiring relevant projects to meet the development needs of Yuanjia village and to ensure the interests of community residents. Under the supervision of the local government, foreign tourism enterprises have established long-term partnerships with Yuanjia village and invested in the large-scale development and construction of Yuanjia village on the basis of preserving the original village landscape and the daily life of the community residents, thus providing a new impetus for the high-quality development of rural tourism in Yuanjia village and accelerating the process of rural revitalisation.

In summary, with the joint efforts of the higher government and the external enterprises, tourism in Yuanjia village has developed rapidly and has quickly become a nationally renowned model village for rural tourism. In this process, the higher government has mainly provided policy and financial and intellectual support for the development of tourism in Yuanjia village from the macro level, while the external enterprises, together with the grassroots organisation of Yuanjia village and the tourism company it led, formed the Scenic Area Operation Centre to manage the tourism activities in a unified manner.

By tracing the development of rural tourism in Yuanjia village, it is easy to see the evolutionary logic of Yuanjia village's tourism-driven rural revitalisation mechanism from "endogenous power" to "exogenous power". Although Yuanjia village has achieved great success in the practice of tourism rural revitalisation, the "Yuanjia village" model is not universally applicable in the Chinese context. This is because Yuanjia village accumulated

rich capital and market experience and a deep community base through the development of the village collective economy prior to the development of rural tourism, and it has a strong ability to transform and develop in the face of external opportunities and risks [86]. As a result, Yuanjia village has been able to dominate the game of rural tourism development with the local government and foreign companies and other relevant stakeholders. However, most of the villages in China that have certain tourism development foundations and conditions lack the original accumulation that Yuanjia village had before developing tourism. After external forces intervene in rural tourism development, these villages are very likely to lose control and dominance over tourism development and rural construction, which is not conducive to the exercise of community residents' subjectivity or to the sustainable development of rural tourism. Therefore, the evolutionary path of Yuanjia village's rural revitalisation model is only applicable to villages with a good foundation for tourism development and strong grassroots organisational leadership.

(2) From 'Exogenous Power' to 'Endogenous Power': Practices in the Surrounding Villages of Yuanjia Village

With the rapid development of rural tourism in Yuanjia village, the limited land, resources, manpower, and other tourism development factors in Yuanjia village could no longer meet the increasing demand for tourism. In this context, in order to give full play to the radiating effect of tourism in Yuanjia Village, and under the guidance of the local government, Yuanjia village put forward a development strategy of "one village leading ten villages" in 2013, whereby the better-developed cooperatives in Yuanjia village offered shares to poor households in the surrounding villages. The stalls were provided free of charge to the poor households in the surrounding villages on the main roads so that they could engage in tourism operations within their means; this helped the poor households to relinquish the status quo of "having physical strength but not being able to do so" and provided them with a stable source of income.

As the ten villages around Yuanjia village generally lack a mature industrial system and supporting facilities, the industry has low-risk resistance and limited blood-making capacity, resulting in a low willingness of the community residents to participate in the village's industry and insufficient endogenous power and self-development capacity, which is the source of sustainable rural development [87]. If the villages lack the power of endogenous development and the ability to develop themselves, rural development will not be sustainable [88] and rural revitalisation will not be realised as scheduled. Therefore, at the beginning of the development strategy of "one village leading ten villages", with the support of the local government, the Yuanjia larger community, including Yuanjia village and the surrounding villages, was established to manage and guide the tourism development and rural revitalisation of the surrounding villages. Although the development model is still "government-led", in the practice of rural revitalisation in the surrounding villages, both the higher government and the Yuanjia community have paid great attention to the stimulation of the endogenous motivation of the residents of the surrounding villages and the cultivation of their self-development ability. On the basis of promoting the construction of infrastructure and the improvement of the village appearance in the surrounding villages, the local and grassroots governments organised propaganda and moral evaluation to guide the villagers in the surrounding villages to gradually form the ideology of wanting to relinquish poverty and to take the initiative to do so, and they carried out technical training based on the actual industries in each village to enhance the villagers' ability to develop themselves. In particular, the Yuanjia community focuses on strengthening the grassroots organisations in the surrounding villages, and through the promotion of the "Yuanjia" experience, the grassroots organisations in the surrounding villages have gradually become stronger. These initiatives have injected sufficient endogenous impetus for poverty alleviation and rural revitalisation in the surrounding villages. As the villagers in the surrounding villages continue to enhance their self-development capacity, the local government has gradually returned the leadership of rural revitalisation to the community residents of the surrounding villages, stimulating the endogenous motivation of the community residents

to develop tourism through tourism empowerment and then realising rural revitalisation mainly through the power of the community residents.

The implementation of the "one village leading ten villages" strategy has proven that it can provide space for the high-quality development of rural tourism in Yuanjia village, and it has also provided a major opportunity for the surrounding villages to develop special agriculture around Yuanjia village tourism. At present, the surrounding villages of Yuanjia village are based on their own resource endowments and have formed a certain scale of local agriculture with distinctive features. Although the surrounding villages of Yuanjia village have not yet been fully revitalised, under the current "government–community" interlocking model, the practice of tourism promoting rural revitalisation is rapidly increasing.

In the Chinese context, government leadership is both a typical feature of China's large-scale poverty alleviation efforts [89] and a typical choice in the early stages of the rural revitalisation in China. Compared to Yuanjia village's evolutionary path from "endogenous power" to "exogenous power" as the driving mechanism for rural revitalisation, the surrounding villages have evolved from "exogenous power" to "endogenous power" as the driving mechanism for rural revitalisation. Compared with the "endogenous power" to "exogenous power" path of rural revitalisation in Yuanjia village, the "exogenous power" to "endogenous power" path of rural revitalisation in the surrounding villages is obviously more in line with the actual development of Chinese villages and has a more universal promotion value at this stage.

In summary, the motivation mechanism and evolutionary logic of tourism promoting rural revitalisation are closely related to the model of tourism promoting rural revitalisation. Among them, the "government-led" model emphasises top-down exogenous impetus, while the "community-led" model emphasises bottom-up endogenous impetus cultivation, while the "government-community" interlocking can effectively stimulate the synergy of endogenous and exogenous forces in rural tourism development and promote the sustainable development of rural tourism and thus sustainably promote rural revitalisation. On this basis, this paper takes Yuanjia village in Liquan County, Shaanxi Province, a model village of tourism promoting rural revitalisation in China, as a typical case. Through the analysis of the practical process of Yuanjia village and its surrounding villages, it is verified that the logic and path of the evolution of the dynamic mechanism from "exogenous power" to "endogenous power" is more suitable for the actual situation of tourism-assisted rural revitalisation in China.

## 5. Discussion and Conclusion

### 5.1. Discussion

This paper first constructed a coupling relationship model between the comprehensive benefits of rural tourism and the general requirements of rural revitalisation, which completes the theoretical premise of the study on tourism promoting rural revitalisation and makes up for the shortcomings of the previous related studies.

In order to theoretically clarify the motivation mechanism of tourism promoting rural revitalisation, this paper constructed a model of the motivation mechanism of tourism promoting rural revitalisation based on the dialectical unity between endogenous and exogenous driving forces. The model revealed the role of endogenous and exogenous power in tourism promoting rural revitalisation, which has been fully confirmed in previous studies [90,91], indicating that both endogenous and exogenous power play an important role in the practice of tourism promoting rural revitalisation in the Chinese context, but the difference lies in the evolutionary logic and path of the motivation mechanism of tourism promoting rural revitalisation.

Based on the theoretical connection between the model and the motivation mechanism of tourism promoting rural revitalisation, this paper conducted an in-depth analysis of the motivation mechanism and the evolutionary logic of tourism promoting rural revitalisation under the government-led and community-led models. Then, by comparing

the advantages and disadvantages of the two models, the paper proposed an innovative model of the evolutionary path of tourism promoting rural revitalisation under the "government–community" interlocking model. Compared with the previous studies that emphasise either the "government-led" model or the "community-led" model [92,93], the "government–community" interlocking model proposed in this study combines both the "government-led" and the "community-led" models. The "government–community" interlocking model proposed in this study effectively combined the advantages of both models, which is more conducive to giving full play to the synergy between endogenous and exogenous power in the practice of tourism promoting rural revitalisation.

Finally, this paper took Yuanjia village in Liquan County, Shaanxi Province, a model village of tourism promoting rural revitalisation in China, as a typical case and verified the rationality of the evolutionary path of the dynamics of the motivation mechanism of tourism promoting rural revitalisation in Yuanjia village and the surrounding villages from the practical level, through an in-depth analysis of the "government-community" interlocking model. Related research also proves that the evolution from "exogenous dynamics" to "endogenous dynamics" is more universal in the practice of tourism promoting rural revitalisation in China [87,88].

The main contributions of our research are as follows. Theoretically, we constructed a model of the motivation mechanism of tourism promoting rural revitalisation, and by comparing the evolutionary logic of the motivation mechanism of tourism promoting rural revitalisation under the two different government-led and community-led models, we innovatively proposed a model of the evolutionary path of the motivation mechanism of tourism promoting rural revitalisation under the model of "government–community" interlocking, which provides a theoretical tool for analysing the interaction mechanism between the endogenous and exogenous power of tourism promoting rural revitalisation. This model is also of theoretical value to the practice of tourism promoting rural revitalisation in other developing countries around the world.

Admittedly, there are a number of limitations to this study. Firstly, the study case has a certain specificity. Yuanjia village is unique in its practice of tourism promoting rural revitalisation. The practice has proved that many villages that lack original tourism accumulation simply copy the "Yuanjia village model", and their practices of tourism promoting rural revitalisation are not successful [94]. In addition, the surrounding villages of Yuanjia village are still at the initial stage of tourism promoting rural revitalisation, and it is not yet possible to accurately assess the extent of the impact of tourism in Yuanjia village on the rural revitalisation of the surrounding villages and the problems that may arise. Finally, in the process of the evolution of the motivation mechanism from "exogenous drive" to "endogenous drive", how to deal with the relationship between "exogenous drive" and "endogenous drive" at the different development stages is still a topic that needs to be studied in detail in the future.

*5.2. Conclusions*

Through a theoretical exploration and a practical verification of the motivation mechanism and evolutionary logic of tourism promoting rural revitalisation, this paper mainly draws the following conclusions. Firstly, there is a natural coupling between the comprehensive benefits of rural tourism and the general requirements of rural revitalisation; secondly, endogenous power is the fundamental force of tourism promoting rural revitalisation, and exogenous power is the supporting force of tourism promoting rural revitalisation. Thirdly, both the government-led and the community-led models can evolve into the "government–community" interlocking model through certain institutional arrangements.

**Author Contributions:** Theoretical analysis, H.W.; data curation, L.P.; model construction, K.B. and H.W.; writing—original draft, H.W.; writing—review and editing, K.B., X.L. and P.M. All authors have read and agreed to the published version of the manuscript.

**Funding:** This research was supported by the National Social Science Foundation of China, Western China (No. 16XGL008, 20XJY018), the National Natural Science Foundation of China (No. 42071186), the China Postdoctoral Science Foundation Project (No. 2017M623097), the Soft Science Research Program Project of the Shaanxi Provincial Department of Science and Technology (No. 2018KRM056), the Research Project on Major Theoretical and Practical Issues in Social Sciences of Shaanxi Province (No. 2018Z076), the Research Project on Culture, Art and Tourism of the Ministry of Culture and Tourism (No. 19DY14), the Research Project on Major Theoretical and Realistic Issues in Philosophy and Social Sciences of Shaanxi Province (No. 2021ND0237), Shaanxi Provincial Decision-making Advisory Committee Research Project, the Specialized Project in Philosophy and Social Science Research in Shaanxi Province (Key Topics of Ecological Space Governance in Shaanxi Province in 2022)(No. 2022HZ1818), Xi'an Social Science Planning Fund Project (No. JG216), and the Project on the Prosperity of Philosophy and Social Sciences of Xi'an University of Science and Technology (No. 2020SZ01).

**Institutional Review Board Statement:** Not applicable.

**Informed Consent Statement:** Not applicable.

**Data Availability Statement:** The data used to support the findings of this study are available from the first author upon request.

**Acknowledgments:** The research has received considerable support, help, and cooperation from the Yuanjia village committee and the villagers in Liquan County, Shaanxi Province, China.

**Conflicts of Interest:** The authors declare no conflict of interest.

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
