# Peer review of "The Motivation Mechanism and Evolutionary Logic of Tourism Promoting Rural Revitalisation: Empirical Evidence from China"

_sustainability, doi:10.3390/su15032336_

Round 1

Reviewer 1 Report

You should restructure your paper (introduction, theoretical background, methodology, results and discussion, conclusions) so that the readers can understand your research better.

You should clearly state the goal(s) of your paper.

Author Response

Dear reviewer

We are the authors of the paper (Manuscript ID: sustainability-2064682).

Thank you very much for your review of the paper, your suggestions helped a lot in revising the paper.

I have revised the paper according to your suggestions.

The following is a revision of the paper.

Suggestion or question 1.You should restructure your paper (introduction, theoretical background, methodology, results and discussion, conclusions) so that the readers can understand your research better.

Response on the revision:Thank you very much for your suggestion. Based on your suggestions, we have restructured the paper, which includes an introduction, literature review, research methodology, research results and analysis, discussion and conclusion.

Suggestion or question 2. You should clearly state the goal(s) of your paper.

Response on the revision: We further clarify the objectives of the study in the final paragraph of the introductory section.

If there are modifications that are not in place, please propose more modifications. We will definitely revise it seriously.

Thanks again for your review!

Kind regards

                          Huizhan Wang, Kai Bai, Lulu Pei, Xinru Lu and Polish Mohanto

                                                    January 16th, 2023

Reviewer 2 Report

The manuscript proposed for review is of particular interest. As a researcher working in the field of sustainable tourism, I accept that it enriches scientific knowledge on the subject. At the same time, I also have some notes and recommendations regarding the:

- Abstract - in my opinion, it would be good to describe the methodology used in detail in the Abstract. At the moment, it is missing, and not only here, but also in the article as a whole. This reduces the overall impression of it, as well as the clear and sound and understanding when reading;

- Introduction - it would be good to expand the text by presenting, in addition to the narrow focus on China and the official doctrines, the general state of the subject - internationally and in China;

- The literature review - it would be good to be restructured. For example, there are separate subsections in the text here that relate to methodological issues;

- Methodology – to be distinguished. Such a section is missing, but it is of particular importance;

- Results – such section is missing. The text goes directly to discussion;

- Conclusion – to be only pointed only Conclusion, omitting the rest of the subtitle;

- Illustrative materials - in general, the figures in the article are well marked in the text. However, it is not clear what/who are their sources - are these the authors' works or are they someone else's results? I suggest paying attention to this matter and making the necessary additions;

I would also recommend reviewing the text and refining the English pronunciation.

“The Motivation Mechanism and Evolutionary Logic of Tourism Promoting Rural Revitalization: Empirical Evidence from China”

Additional comments

The manuscript proposed for review is of particular interest. As a researcher working in the field of sustainable tourism, I accept that it enriches scientific knowledge on the subject. At the same time, I also have some notes and recommendations regarding the:

- Abstract - in my opinion, it would be good to describe the methodology used in detail in the Abstract. At the moment, it is missing, and not only here, but also in the article as a whole. This reduces the overall impression of it, as well as the clear and sound and understanding when reading;

An additional highlight is the clarification of the last sentence, where the authors speak directly on their own behalf, which is not entirely acceptable for a scientific text.

- Introduction - it would be good to expand the text by presenting, in addition to the narrow focus on China and the official doctrines, the general state of the subject - internationally and in China;

Again, I would emphasize the strong inclination of the text towards the theory and doctrine of China, in the implementation of the state strategic plans, although both the Abstract and the Introduction itself, at the end of the section – the last paragraph - talk about the presentation of different models. Thus, it is not clear which models are analysed and what about they are: for rural development, for tourist promotion or rural revitalization or otherwise…

Actually, that was the point of my original comment. To provoke the authors to ask themselves and explain what exactly they mean when they talk about "different models".

In the spirit of the additional comments requested of me, of course, I can offer one more thing: fig. 1 of the Introduction to be moved further down in the text. It can be presented, for example, in connection with the literature review or in the general discussion

- The literature review - it would be good to be restructured. For example, there are separate subsections in the text here that relate to methodological issues;

It would be good to refine the literature review - on the one hand, in relation to the practice in China, and on the other hand, in relation to the possibilities of revitalizing rural areas with the help of tourism and tourism promotion. In this sense, it would be good to present the perspectives of authors outside the China and Asia region.

- Methodology – to be distinguished. Such a section is missing, but it is of particular importance;

A separate part presenting the research methodology is mandatory. Its addition requires further restructuring of the text, as the methodological issues are currently presented in a very messy manner.

- Results – such section is missing. The text goes directly to discussion;

As an approach, a kind of case study was used. However, there is no explanation in this regard and it is good to supplement it by making the necessary justification.

In this way, the presentation of the results will be more organized and clearer. Moreover, the selection of a specific example can serve as a good practice against which to organize the tourism promotion for rural revitalization in other places with potential.

- Conclusion – to be only pointed only Conclusion, omitting the rest of the subtitle;

Regarding the Conclusion, the comments are clear. The addition "political implications" is not necessary, moreover, the text again refers to the decisions at the governmental level.

In this sense, there can be no political implications. Rather, the authors most likely are looking for the possible connections and projections of these solutions from 2022, which are actually of particular importance for the territory of China.

- Illustrative materials - in general, the figures in the article are well marked in the text. However, it is not clear what/who are their sources - are these the authors' works or are they someone else's results? I suggest paying attention to this matter and making the necessary additions.

I would also recommend reviewing the text and refining the English pronunciation.

My overall impression of the proposed manuscript is positive. I would recommend to be published after removing the indicated weaknesses (of course at the discretion of the authors). I believe that it enriches theory and practice with concrete examples from East Asia and more specifically China.

It would definitely be better to restructure the manuscript and outline the chosen methodology and research results. This would contribute to a better understanding of the text and what the authors have achieved as a study and conclusions.

Author Response

Dear reviewer

We are the authors of the paper (Manuscript ID: sustainability-2064682).

Thank you very much for your review of the paper, your suggestions helped a lot in revising the paper.

I have revised the paper according to your suggestions.

The following is a revision of the paper.

Suggestion or question 1.-Abstract - in my opinion, it would be good to describe the methodology used in detail in the Abstract. At the moment, it is missing, and not only here, but also in the article as a whole. This reduces the overall impression of it, as well as the clear and sound and understanding when reading;

——An additional highlight is the clarification of the last sentence, where the authors speak directly on their own behalf, which is not entirely acceptable for a scientific text.

Response on the revision:We have added a description of the research methodology to the abstract and a third point "Research Methodology" to the body of the text. And the last sentence replaces the statement.

Suggestion or question 2. - Introduction - it would be good to expand the text by presenting, in addition to the narrow focus on China and the official doctrines, the general state of the subject - internationally and in China;

——Again, I would emphasize the strong inclination of the text towards the theory and doctrine of China, in the implementation of the state strategic plans, although both the Abstract and the Introduction itself, at the end of the section–the last paragraph - talk about the presentation of different models. Thus, it is not clear which models are analysed and what about they are: for rural development, for tourist promotion or rural revitalization or otherwise…

——Actually, that was the point of my original comment. To provoke the authors to ask themselves and explain what exactly they mean when they talk about "different models".

Response on the revision: Firstly, in the introduction section, the thesis briefly adds to the international exploration of rural revitalisation in practice; Secondly, in the introductory part, the characteristics of the motivation mechanism of tourism promoting rural revitalisation under different tourism development models are briefly explained; Thirdly, Figure 1 and related contents in the introduction are reorganized into the second part, i.e. "Literature Review".

Suggestion or question 3. - The literature review - it would be good to be restructured. For example, there are separate subsections in the text here that relate to methodological issues;

——It would be good to refine the literature review - on the one hand, in relation to the practice in China, and on the other hand, in relation to the possibilities of revitalizing rural areas with the help of tourism and tourism promotion. In this sense, it would be good to present the perspectives of authors outside the China and Asia region.

Response on the revision: We have restructured the literature review by adding a review of 'research methods'. At the same time, we have added the views of some foreign scholars in the field of rural revitalisation research.

Suggestion or question 4. - Methodology – to be distinguished. Such a section is missing, but it is of particular importance;

——A separate part presenting the research methodology is mandatory. Its addition requires further restructuring of the text, as the methodological issues are currently presented in a very messy manner.

Response on the revision: We have added a third section "Research Methodology" to the original text , which explains the research methodology of the thesis.

Suggestion or question 5. - Results – such section is missing. The text goes directly to discussion;

——As an approach, a kind of case study was used. However, there is no explanation in this regard and it is good to supplement it by making the necessary justification.

——In this way, the presentation of the results will be more organized and clearer. Moreover, the selection of a specific example can serve as a good practice against which to organize the tourism promotion for rural revitalization in other places with potential.

Response on the revision: We have added a paragraph summarising the findings and cases at the end of Section 4 (Research results and analysis).

Suggestion or question 6. - Conclusion – to be only pointed only Conclusion, omitting the rest of the subtitle;

——Regarding the Conclusion, the comments are clear. The addition "political implications" is not necessary, moreover, the text again refers to the decisions at the governmental level.

——In this sense, there can be no political implications. Rather, the authors most likely are looking for the possible connections and projections of these solutions from 2022, which are actually of particular importance for the territory of China.

Response on the revision: We deleted the policy implications and omitted the rest of the subtitle.

Suggestion or question 7.- Illustrative materials - in general, the figures in the article are well marked in the text. However, it is not clear what/who are their sources - are these the authors' works or are they someone else's results? I suggest paying attention to this matter and making the necessary additions;

——My overall impression of the proposed manuscript is positive. I would recommend to be published after removing the indicated weaknesses (of course at the discretion of the authors). I believe that it enriches theory and practice with concrete examples from East Asia and more specifically China.

——It would definitely be better to restructure the manuscript and outline the chosen methodology and research results. This would contribute to a better understanding of the text and what the authors have achieved as a study and conclusions.

Response on the revision: Firstly,We have supplemented the relevant data sources on the case sites with additional references [84]. Secondly,we have restructured the paper, which includes an introduction, literature review, research methodology, research results and analysis, discussion and conclusion.

Suggestion or question 8.I would also recommend reviewing the text and refining the English pronunciation.

Response on the revision: We have carefully checked the pronunciation throughout the text, for example by changing "-ze" to "-se". See the full text for details.

If there are modifications that are not in place, please propose more modifications. We will definitely revise it seriously.

Thanks again for your review!

Kind regards

                          Huizhan Wang, Kai Bai, Lulu Pei, Xinru Lu and Polish Mohanto

                                                    January 16th, 2023

Reviewer 3 Report

I absolutely love the topic, methodology, contributions, research and conclusion. English is not my native language, so I can say that it is correct as far as I am concerned.

Author Response

Thank you very much for your comments on this article.
